

# Inheritance of heat tolerance in perennial ryegrass (*Lolium perenne*, Poaceae): evidence from progeny array analysis

Wagdi S. Soliman[1], Ahmed M. Abbas[2,3], Stephen J. Novak[4], Masahiro Fujimori[5], Kazuhiro Tase[6] and Shu-ichi Sugiyama[7]

[1] Department of Horticulture, Faculty of Agriculture and Natural Resources, Aswan University, Aswan, Egypt
[2] Department of Biology, College of Science, King Khalid University, Abha, Saudi Arabia
[3] Department of Botany and Microbiology, Faculty of Science, South Valley University, Qena, Egypt
[4] Department of Biological Sciences, Boise State University, Boise, ID, United States of America
[5] Yamanashi Dairy Agricultural Station, Yamanashi, Japan
[6] National Agricultural Research Center for Hokkaido Region, Sapporo, Japan
[7] Hirosaki University, Hirosaki, Japan

Corresponding author
Wagdi S. Soliman,
wagdi79@agr.aswu.edu.eg

## ABSTRACT

**Background**. Heat stress is considered one of the most important environmental factors influencing plant physiology, growth, development, and reproductive output. The occurrence and damage caused by heat stress will likely increase with global climate change. Thus, there is an urgent need to better understand the genetic basis of heat tolerance, especially in cool season plants.

**Materials and Methods**. In this study, we assessed the inheritance of heat tolerance in perennial ryegrass (*Lolium perenne* L. subspecies *perenne*), a cool season grass, through a comparison of two parental cultivars with their offspring. We crossed plants of a heat tolerant cultivar (Kangaroo Valley) with plants of a heat sensitive cultivar (Norlea), to generate 72 F1 hybrid progeny arrays. Both parents and their progeny were then exposed to heat stress for 40 days, and their photosynthetic performance (Fv/Fm values) and leaf $H_2O_2$ content were measured.

**Results**. As expected, Kangaroo Valley had significantly higher Fv/Fm values and significantly lower $H_2O_2$ concentrations than Norlea. For the F1 progeny arrays, values of Fv/Fm decreased gradually with increasing exposure to heat stress, while the content of $H_2O_2$ increased. The progeny had a wide distribution of Fv/Fm and $H_2O_2$ values at 40 days of heat stress. Approximately 95% of the 72 F1 progeny arrays had Fv/Fm values that were equal to or intermediate to the values of the two parental cultivars and 68% of the progeny arrays had $H_2O_2$ concentrations equal to or intermediate to their two parents.

**Conclusion**. Results of this study indicate considerable additive genetic variation for heat tolerance among the 72 progeny arrays generated from these crosses, and such diversity can be used to improve heat tolerance in perennial ryegrass cultivars. Our findings point to the benefits of combining physiological measurements within a genetic framework to assess the inheritance of heat tolerance, a complex plant response.

## INTRODUCTION

Based on the various habitats they occupy, plants require certain environmental conditions to maintain the abundance and persistence of their populations (*Harper, 1977*). During their lifetime, however, most plants experience abiotic stress when exposed to unfavorable chemical and physical environmental conditions such as heavy metals, high salinity, excessive solar radiation, freezing temperatures, severe drought, and extremely high temperatures (*Nilsen & Orcutt, 1996*). Of these stressors, drought and heat stress are among the two most important environmental factors influencing plant physiology, growth, development, and reproductive output (i.e., yield) (*Jiang & Huang, 2001*; *Prasad, Staggenborg & Ristic, 2008*; *Jespersen, Belanger & Huang, 2017*).

According to *Wahid et al. (2007)*, heat stress (or heat shock) in plants occurs when temperatures rise above a threshold level for sufficient time to result in irreversible damage to plant growth and development. Although heat stress usually occurs with an increase in temperature of 10−15 °C above ambient; heat stress is also influenced by the intensity, duration, and rate of increase in temperature (*Wahid et al., 2012*). Thus, heat stress in plants can occur on a daily or seasonal basis and can vary from year-to-year. In addition, the occurrence and damage caused by heat stress will likely increase with global climate change (*Walter et al., 2013*; *Bita & Gerats, 2013*). Due to human activities, substantial increases in the concentration of greenhouse gases are occurring, and global air temperatures are predicted to increase 1−4.5 °C above the current level by 2100, depending on different carbon emission scenarios (*Rogelj, Meinhausen & Knutti, 2012*; *IPCC, 2019*). Moreover, human-caused climate change is also associated with extreme climate events such as precipitation extremes, flooding, frosts, drought, and excessive heat (*Niu et al., 2014*; *Stott, 2016*). Thus, future climate change is expected to cause serious damage to the growth and yield of native plants and crop plants, especially $C_3$ plants and crops (*Lobell & Field, 2007*; *Wahid et al., 2012*).

Exposure of plants to excessively high temperatures can result in a range of complex responses from molecular and cellular, to whole plant levels (*Baniwal et al., 2004*; *Kotak et al., 2007*; *Wahid et al., 2012*; *Prasad, Staggenborg & Ristic, 2008*; *Mittler, Finka & Goloubinoff, 2011*; *Soliman et al., 2011*). Once leaf temperatures rise above a threshold level (35 to 40 °C for most plants), protein denaturation and loss of cell membrane fluidity begins to take place and cell damage and programmed cell death may occur (*Huang & Xu, 2008*; *Horvath et al., 2012*). Depending on the intensity and duration of exposure to high temperatures, plant tissue type, and phenological stage, heat stress in plants can induce the following responses: (1) loss of cell water content, (2) reduced photosynthetic activity, (3) oxidative stress, (4) scorching of tissues and premature leaf senescence and abscission, (5) reduced growth rates through inhibition of shoot and root growth, (6) damage or alteration of floral (reproductive) tissues, and (7) reduced seed number and quality (see Figure 1 in *Hasanuzzaman et al., 2013*).

With heat stress, reductions in photosynthetic activity and efficiency may take place because high temperatures can lead to the dissociation or inhibition of oxygen evolving complexes (OEC) and reduce the activity of photosystem II (PSII) (*Wahid et al., 2007*).

Photosynthetic performance during heat stress can be quantified by measuring chlorophyll fluorescence parameters (*Baker & Oxborough, 2004*; *Rosyara et al., 2010*). One such parameter is Fv/Fm, which is calculated as the ratio between variable fluorescence (Fv = Fm - Fo) and maximum fluorescence (Fm). Exposure to high temperatures can also induce oxidative stress in plants by uncoupling enzymes and metabolic pathways which generates reactive oxygen species (ROS) that can damage multiple cellular organelles and physiological processes (*Locato et al., 2008*; *Soliman et al., 2011*; *Soliman et al., 2012*; *Hasanuzzaman et al., 2013*). Reactive oxygen species (ROS) include singlet oxygen, superoxide radical, hydroxyl radical, and hydrogen peroxide ($H_2O_2$). Hydrogen peroxide is the most stable ROS and its adverse effects in plants include membrane lipid peroxidation, toxicity, and cell death. Interestingly, recent studies have highlighted the important role of $H_2O_2$ as a signaling molecule in plants, triggering tolerance responses to abiotic stresses (*Suzuki et al., 2012*; *Baxter, Mittler & Suzuki, 2014*).

Plants are sessile organisms that are less likely to evade abiotic stressors. Thus, they have developed several mechanisms for mitigating and surviving heat stress (see Figure 4 in *Hasanuzzaman et al., 2013*). These mechanisms include short-term avoidance or acclimation mechanisms and long-term phenological and morphological adaptive traits such as early maturation, enhanced root density and depth, changing leaf orientation and leaf rolling, transpirational cooling, and/or alteration of membrane lipid compositions (*Wahid et al., 2007*; *Wahid et al., 2012*; *Prasad, Staggenborg & Ristic, 2008*; *Hasanuzzaman et al., 2013*; *Jespersen, Belanger & Huang, 2017*). Additionally, plants have developed molecular, cellular, and physiological adaptations for tolerating heat stress (*Wahid et al., 2012*; *Hasanuzzaman et al., 2013*). These include signaling cascades and regulation of gene expression by transcription factors (*Yang et al., 2014*; *Ohama et al., 2016*; *Jespersen, Belanger & Huang, 2017*), expression of heat shock proteins (HSPs) and molecular chaperones (*Horvath et al., 2012*; *Davies et al., 2018*), enzymatic and non-enzymatic antioxidant defense to prevent the harmful effects of ROS (*Gulen & Eris, 2004*), and the production of osmo-protectants or compatible solutes (*Wahid et al., 2007*; *Hasanuzzaman et al., 2013*). Clearly, heat tolerance in plants is controlled by a complex set of many genes, interacting mechanisms, and phenotypic traits, and not just a single gene, mechanism, or trait (*Erdayani et al., 2020*).

Perennial ryegrass (*Lolium perenne* L. subspecies *perenne*) is a cool season (C$_3$), perennial grass that has a caespitose (bunch) growth form and can grow to a height of approximately 10–90 cm. Perennial ryegrass originated in the Middle East, and then dispersed across Europe and North Africa with the spread of agriculture (*Balfourier, Imbert & Charmet, 2000*). Perennial ryegrass has subsequently been introduced around the globe and is considered a weed, or an invasive species, in natural communities in many regions. It is also one of the most common pasture grasses in temperate climatic regions where it is used as a forage grass for livestock and for hay production. In addition, it is widely used as a turf grass (*Bolaric et al., 2005a*; *Bolaric et al., 2005b*; *Wang et al., 2009*), and for restoration and conservation seedings. Perennial ryegrass is naturally a diploid species (2n = 2x = 14) (*Bolaric et al., 2005b*; *Wang et al., 2009*), but tetraploid (2n = 4x = 28) cultivars have also been developed (*Nair, 2004*). The grass has a two-locus self-incompatibility system,

which leads to an obligately outcrossing mating system (*Cornish, Hayward & Lawrence, 1979*). This mating system ensures outbreeding among individuals and high amounts of genetic diversity within naturally occurring populations (*Bolaric et al., 2005a*; *Bolaric et al., 2005b*; *Wang et al., 2009*). Many cultivars of perennial ryegrass however are derived from a limited pool of foundational clones; and such cultivars typically exhibit a limited amount of genetic variation (*Guthridge et al., 2001*).

Because perennial ryegrass is a cool season grass of temperate regions, it is generally considered to be sensitive to heat stress (*Li, Jannasch & Jiang, 2020*); although heat tolerant cultivars have been developed (*Wilkins, 1991*). In addition, because the grass is widely cultivated and has high economic value, experiments assessing heat stress in perennial ryegrass, especially comparisons of heat tolerant and heat sensitive cultivars, have been conducted (e.g., *Wehner & Watschke, 1981*; *Jiang & Huang, 2001*; *Zhou & Abaraha, 2007*; *Wang et al., 2017*; *Sun et al., 2020*; *Li, Jannasch & Jiang, 2020*). For example, in a previous study we reported that a heat tolerant cultivar of perennial ryegrass (Yatugadake-24) exhibited significantly higher photosynthetic performance (i.e., the plants had higher Fv/Fm values) and lower leaf $H_2O_2$ content, compared to a heat sensitive cultivar (Norlea) (*Soliman et al., 2011*). In another study, *Soliman et al. (2012)* exposed 25 diploid and tetraploid cultivars of perennial ryegrass to prolonged heat stress and found that tetraploid cultivars had lower $H_2O_2$ content and experienced less oxidative stress than diploid cultivars. Taken together, these studies indicate considerable genetic variation in heat tolerance among perennial ryegrass cultivars and cytotypes. Yet, to the best of our knowledge, we are not aware of any assessment of the genetic basis of heat tolerance in perennial ryegrass.

In this study, we assessed the inheritance of heat tolerance in perennial ryegrass through a direct comparison of parental cultivars with their offspring, through progeny array analysis. This was accomplished by crossing plants of a heat tolerant cultivar of perennial ryegrass with plants of a heat sensitive cultivar, to generate multiple F1 progeny arrays. Both parents and their progeny were then exposed to long-term heat stress, and their photosynthetic performance and leaf $H_2O_2$ concentrations were measured. In addition, several leaf growth parameters were measured before the imposition of heat stress. The specific goals of this research were to, (1) quantify the level of heat tolerance in the two parental cultivars, (2) determine variation in heat tolerance among the F1 progeny, and (3) compare the level of heat tolerance of the parents with their F1 progeny to assess the inheritance of this complex and important plant response. Results of this study will improve our understanding of the genetic basis of heat tolerance in perennial ryegrass, assist in estimating the heritability of this trait, and aid in the identification and selection of plants with even higher levels of heat tolerance for use in plant breeding programs.

## MATERIALS AN METHODS

### Plant material

Plants of two diploid perennial ryegrass cultivars were used as the parents in this study. Kangaroo Valley (strain K7) is a heat tolerant cultivar developed in New South Wales,

Australia, that is well-suited to dry, hot regions (*Wilkins, 1991*; *Blumenthal et al., 1996*) and Norlea (strain N4) is a heat sensitive cultivar developed in Canada (*Soliman et al., 2011*; *Soliman et al., 2012*). Based on the breeding programs that developed them, both strains of the two cultivars exhibit limited genetic diversity (*Blumenthal et al., 1996*; *Soliman et al., 2011*).

Flowers of each of these two cultivars were crossed through hand-pollination, after they were emasculated. These crosses were performed using 72 different pairs of the two cultivars. An individual of the Kangaroo Valley cultivar was always used as the pollen donor and an individual of the Norlea cultivar was always used as the maternal parent. Sufficient hand-pollinations were conducted to generate at least six seeds from each pair of the two cultivars; thus, 72 full-sib, F1 hybrid progeny arrays, each consisting of six seeds, were generated and employed in our experiment design. Crosses were conducted at the Yamanashi Dairy Experimental Station, Yamanashi, Japan.

## Heat stress treatment

Our heat stress experiment was conducted using the procedures described by *Soliman et al. (2012)*. The seeds/seedlings of the two perennial ryegrass strains (K7 and N4) used in the heat stress experiments were not the same individuals used to generate the progeny arrays; but because these two strains have limited genetic diversity, seeds of these two cultivars are genetically uniform. Six seeds (replicates) of each of the two parental cultivars and six seeds from each of the 72 F1 progeny arrays were germinated on wet filter paper in petri dishes. The grass does not require any other treatments to achieve high rates of germination. Seedlings were transplanted into pots (two seeds per pot), 7.5 cm in diameter and eight cm deep, with a sandy loam potting soil containing 0.35 g of N, P2O5, and K2O for every kilogram of soil (N-P-K ratio of 2:1:2). The seedlings were grown in a controlled growth chamber with day/night temperatures of 23/16 °C, a 16h/8 h day/night photoperiod (from 4:00–20:00 h), with photon flux density of 250 $\mu$mol m$^{-2}$ s$^{-1}$, and a constant relative humidity of 70%. Forty days after transplanting, all the plants were exposed to 30 °C for 3 days for acclimation to a higher temperature, after which the plants were exposed to heat treatments (36/30 °C, day/night temperatures) for 40 day. Plants were watered daily to avoid water (drought) stress. The heat stress experiment was set up in a randomized complete block design, with six replicates for each of the two parental cultivars and each of the 72 F1 progeny arrays.

## Leaf growth traits

Leaf growth traits were measured prior to the imposition of heat stress. For each seedling, the second fully matured leaf was selected for measurements. Specific leaf area (SLA) and its components were determined according to the method of *Witkowski & Lamont (1991)*. Specific leaf area is calculated as the ratio of leaf area (LA) to leaf dry mass (LDM). Other leaf measurements included leaf water content (LWC), leaf thickness (LT), and leaf density (LD). Before the start of the heat stress treatment, the second mature leaf from each plant was harvested, its fresh weight was recorded, and it was immediately soaked in a 50 ml flask filled with water to perform measurements with leaves at full turgor. An image of each leaf

was then digitally recorded using an optical scanner (D660U, Canon, Tokyo, Japan). The leaf area was calculated using Image J software (version 1.6, National Institutes of Health). The leaves were then oven dried at 80 °C for two days, and their dry weights were recorded. Leaf thickness was determined using microscopic observation of leaf transverse sections using MICROM (HM400R, Walldorf, Germany) as previously described (*Soliman et al., 2012*). Leaf density (mg/cm$^{-3}$), or dry matter concentration, was calculated by dividing leaf dry mass by leaf volume. Leaf volume was determined as the product of leaf area and mean leaf thickness.

## Photosynthetic performance

Chlorophyll fluorescence (Fv/Fm) values were measured before the initiation of heat acclimation and at 10-day intervals thereafter. Individual seedlings were maintained in the dark for 20 min for dark adaptation and then the minimum ($F_0$) and maximal (Fm) levels of fluorescence were measured three times for each individual using a portable photosynthesis measuring system (LI-6400, Li-Cor, Lincoln, Nebraska, USA). Fv/Fm provides an estimate of the maximum quantum yield of PSII (*Butler, 1978*; *Zhou et al., 2015*); where heat tolerant plants typically exhibit higher Fv/Fm values (i.e., they have higher photosynthetic performance) than heat sensitive plants.

## Oxidative stress

Heat sensitive plants experience greater oxidative stress than heat tolerant plants because plants that are heat sensitive (i.e., experiencing heat stress) produce higher concentrations of $H_2O_2$ than those that are heat tolerant. Hydrogen peroxide ($H_2O_2$) concentration values were determined according to the methods described by *Soliman et al. (2011)*. There were two $H_2O_2$ measurement periods; before the imposition of heat stress and at 40 days of exposure to heat stress. Hydrogen peroxide ($H_2O_2$) content of leaves was measured using a modified version of the ferrous ammonium sulphate/xylenol orange (eFOX) method described by *Cheeseman (2006)* and *Queval et al. (2008)*. Leaf extracts were obtained by grinding 50 mg of leaf tissue, first in liquid nitrogen and then in 500 µL of 0.1 M potassium phosphate buffer (pH 6.5) containing 5 mM NaN$_3$. Extracts were centrifuged at 10,000 rpm (8,385 g) at 5 °C for 5 min. For every 200 µL of the extract, five mL of the solution containing 250 µM ferrous ammonium sulphate, 100 µM sorbitol, 100 µM xylenol orange, 1% ethanol, and 25 mM $H_2SO_4$ were added. The assay consisted of measuring the difference in absorbance between 550 nm and 800 nm, after 15 min, with a spectrophotometer.

## Statistical analyses

Analysis of variance (ANOVA) was used to test for significant differences between the two parental cultivars (six replicates per cultivar) and among the progeny for leaf growth traits; and among the progeny for Fv/Fm values and $H_2O_2$ content (six replicates for each of the 72 F1 hybrid progeny arrays), at different days of exposure to heat stress. Because the same plants were used to measure Fv/Fm values over time, and these data were not independent of each other, we used one-way repeated measures multivariate analysis of variance (MANOVA) to test whether the two cultivars were significantly different. We used a $t$-test to test for significant differences in the $H_2O_2$ content between two cultivars

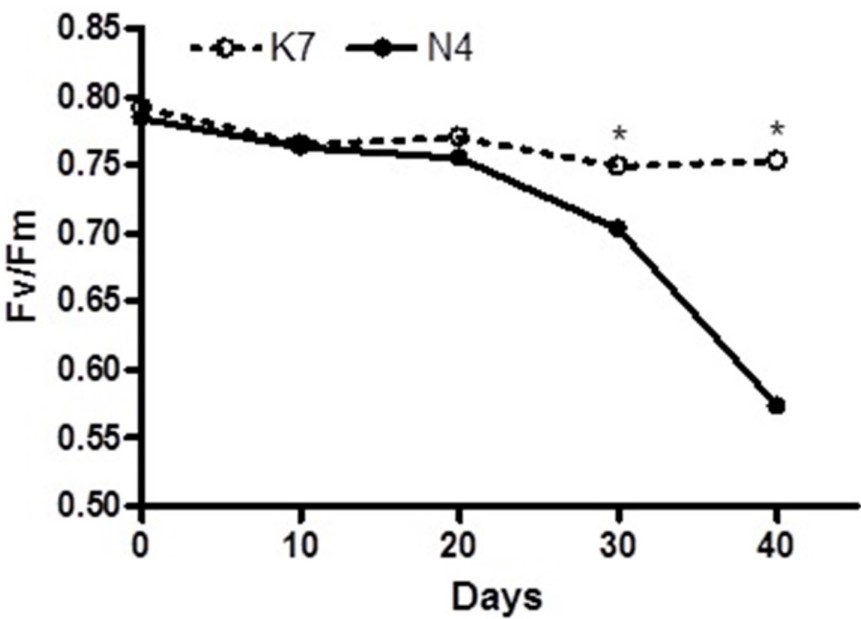

**Figure 1 Temporal changes in chlorophyll florescence (Fv/Fm) values for the two parental cultivars after imposition of heat stress.** Each of the two parental cultivars consisted of six replicates. (●) heat sensitive Norlea (N4) and (○) heat tolerant Kangaroo Valley (K7). An asterisk (*) represents the level of statistical significance at $P < 0.05$.

before and at 40 days of heat stress. A random-effects regression model was used to assess the relationship between leaf growth traits and Fv/Fm values and $H_2O_2$ content for the 72 F1 progeny arrays at 40 days after the imposition of heat stress, with leaf traits as fixed effects and Fv/Fm values and $H_2O_2$ content as random variables. Another random-effects regression model was used to assess the relationship between Fv/Fm values and $H_2O_2$ content for the 72 F1 progeny arrays at 40 days after the imposition of heat stress, with Fv/Fm values and $H_2O_2$ content as random variables. All statistical analyses were carried out using JMP (ver 4. SAS Institute, Cary, NC, USA).

## RESULTS

### Prior to heat stress treatment

Before the imposition of heat stress, there were no statistically significant differences in chlorophyll fluorescence (Fv/Fm) values and hydrogen peroxide ($H_2O_2$) content for the two parental cultivars (Figs. 1 and 2).

Prior to experiencing heat stress, the two parental cultivars did not exhibit significant differences for three of five leaf growth traits and there were significant differences in two traits, leaf water content and leaf thickness (Fig. 3 and Table 1). The two exceptions to this pattern were leaf water content and leaf thickness. Conversely, significant phenotypic variation was observed for all leaf growth traits among the 72 progeny arrays (Table 1), and they exhibited a normal distribution for all five leaf growth traits (Fig. 3). For instance, 71 of 72 (98.6%) progeny arrays had leaf area values that were equal to or greater than the values

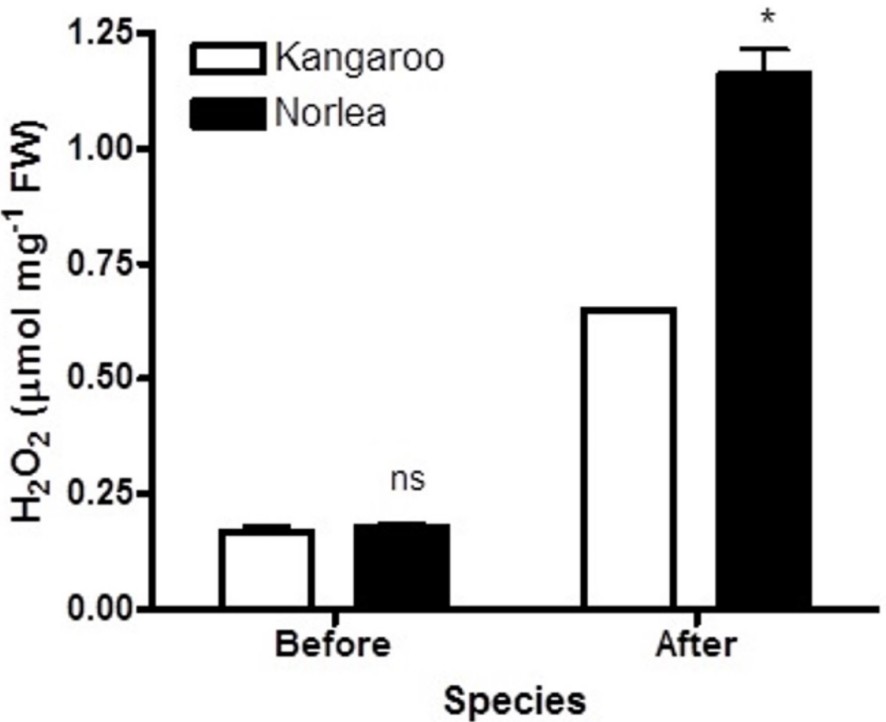

* represents the difference at 0.1%

**Figure 2** **Hydrogen peroxide ($H_2O_2$) content before and after the imposition of heat stress in the two parental cultivars. Each of the two parental cultivars consisted of six replicates.** An asterisk (*) represents the level of statistical significance at $P < 0.01$.

of their two parents (Fig. 3A). However, for leaf water content, 60 of 72 (83%) progeny arrays had values that were equal to or intermediate to the values of the two parental cultivars (Fig. 3C). There was no relationship (no significant correlations) between the leaf growth traits and Fv/Fm values and $H_2O_2$ content at 40 days after the imposition of heat stress (data not shown).

## Response to heat stress: parental cultivars

The two parental cultivars did not exhibit statistically significant differences in Fv/Fm values for the first three measurement periods, at 0 d and after the imposition of heat stress (10 d and 20 d). However, the two parents did show significant differences in their Fv/Fm values at 30 d and 40 d of heat stress, with the Kangaroo Valley cultivar having significantly higher values (i.e., higher photosynthetic performance) (Figs. 1 and 4A). At 40 d of heat stress, both parental cultivars had higher $H_2O_2$ values compared to before the imposition of stress. At 40 d of exposure to heat stress, Norlea, the heat-sensitive cultivar, had significantly higher $H_2O_2$ content than the Kangaroo Valley cultivar (Figs. 2 and 4B). This result indicates that the Norlea cultivar experienced more oxidative stress under these conditions.

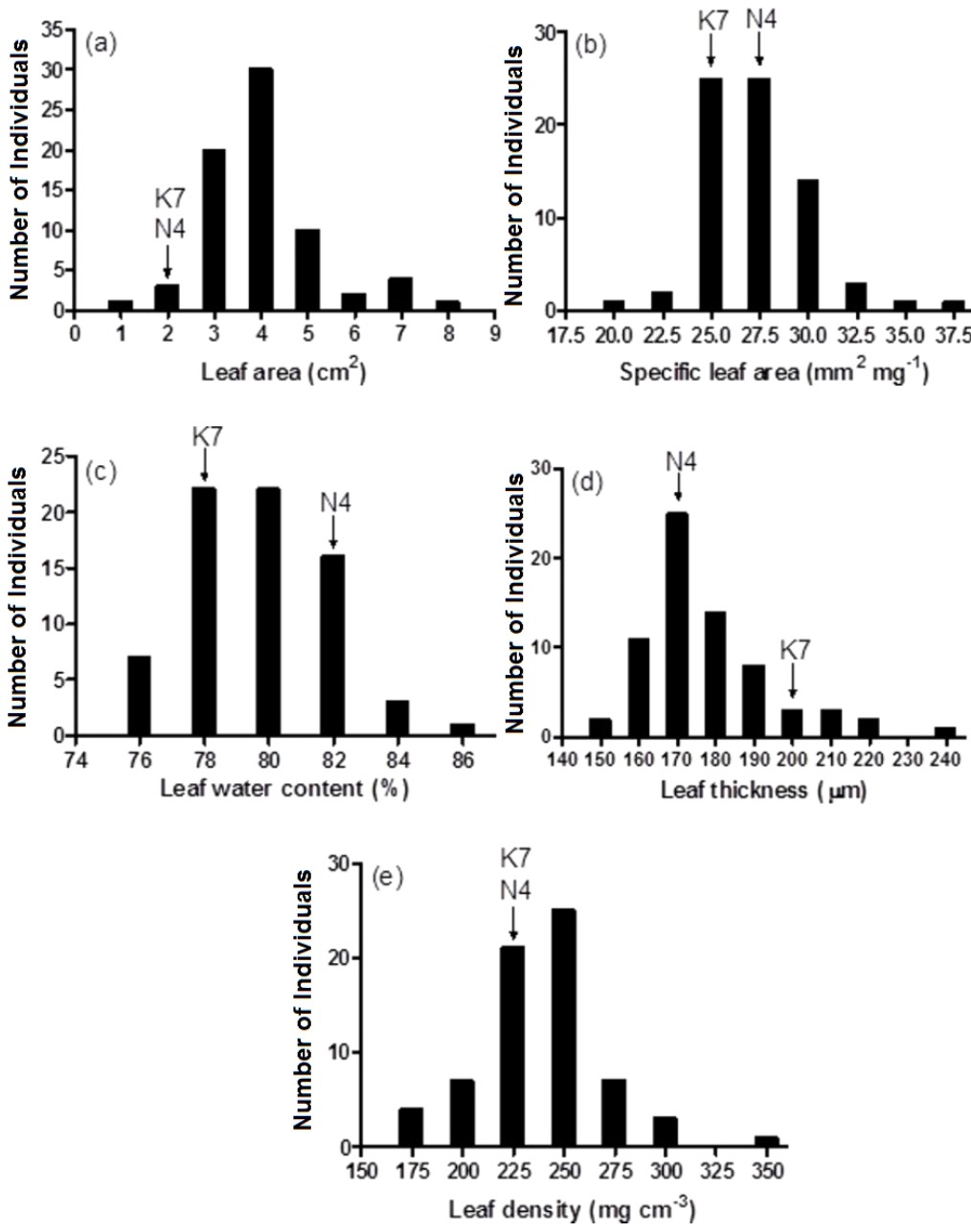

**Figure 3 Frequency distributions of leaf area (A), specific leaf area (B), leaf water content (C), leaf thickness (D) and leaf density (E) for the 72 F1 progeny arrays and the two parental cultivars.** (K7) heat tolerant Kangaroo Valley and (N4) heat sensitive Norlea. Each progeny array and each of the two parental cultivars consisted of six replicates.

## Response to heat stress: progeny arrays

Fv/Fm values for the 72 progeny arrays were significantly different for all time periods measured (0 d to 40 d) (Table 2). Fv/Fm values of the progeny arrays decreased gradually with increased duration of heat stress (Table 2), and a broad distribution of Fv/Fm values

**Table 1  Analysis of variance (ANOVA) for leaf growth traits for the two parental cultivars and the 72 F1 progeny arrays prior to the imposition of heat stress.**

| Leaf growth traits | Parental cultivars | | | 72 F1 progeny | |
|---|---|---|---|---|---|
| | Norlea | Kangaroo | F value | Range | F value |
| Leaf area ($cm^2$) | 1.72 | 1.81 | 0.07 [ns] | 0.87~7.44 | 7.48[***] |
| Specific leaf area ($mm^2$ $mg^{-1}$) | 26.96 | 24.43 | 1.01[ns] | 19.0~35.4 | 3.38[***] |
| Leaf water content (%) | 81.3 | 77.5 | 8.96[*] | 74.5~85.2 | 4.76[***] |
| Leaf thickness ($\mu$m) | 169 | 198 | 7.75[*] | 141~242 | 5.68[***] |
| Leaf density ($mg\ cm^{-3}$) | 221 | 211 | 0.27[ns] | 153~328 | 7.93[***] |

Notes.
Each of the two parental cultivars and each progeny array consisted of six replicates.
[*, ***] indicate the level of statistical significance at $P < 0.05$ and $P < 0.001$, respectively.
[ns] indicates no statistical differences for the two parental cultivars for three leaf growth traits.

was observed at 40 d (Fig. 4A). Sixty-nine of 72 (approximately 95%) of the progeny arrays had Fv/Fm values that were equal to or intermediate to the values of the two parental cultivars.

The progeny had statistically significant variation in $H_2O_2$ values both prior to the imposition of heat stress (at 0 d) and at 40 days after the imposition of heat stress (at 40 d). Additionally, the progeny arrays undergoing heat stress experienced an increase in their $H_2O_2$ content (Table 2, Fig. 4B). At 40 days after the imposition of heat stress, 49 of 72 (68%) of the progeny arrays had $H_2O_2$ values that were equal to or intermediate to the two parental cultivars.

At 40 days of exposure to heat stress, the two cultivars and the 72 progeny arrays exhibited a significant inverse relationship (r = −0.54) between Fv/Fm values and $H_2O_2$ values (Fig. 5). Most of the data points for the progeny arrays in Fig. 5 cluster near the Kangaroo Valley cultivar, which served as the paternal parent in the crosses that produced these progeny.

## DISCUSSION

Perennial ryegrass is one of the most common pasture and turf grasses in temperate climate regions around the globe (*Bolaric et al., 2005a*; *Bolaric et al., 2005b*; *Wang et al., 2009*). Because it is a cool-season grass, it is thought to be sensitive to heat stress (*Li, Jannasch & Jiang, 2020*); however, plant breeders have also developed heat tolerant cultivars (*Wilkins, 1991*). In addition, because many strains of these cultivars are derived from a limited number of individuals, and possess limited genetic diversity (*Guthridge et al., 2001*), they can function similarly to inbred lines. Thus, features of the two perennial ryegrass cultivars we used were essential in designing our study. First, the Kangaroo Valley cultivar is heat tolerant and Norlea is heat sensitive; therefore, these two cultivars are genetically and phenotypically distinct. Second, different seeds of each of the two strains are genetically (and phenotypically) uniform, thus we could reliably use different seeds of each cultivar to generate the 72 progeny arrays and in the heat stress experiment.

The results of the current study are generally consistent with others that have assessed photosynthetic performance and oxidative stress with heat stress in heat tolerant and

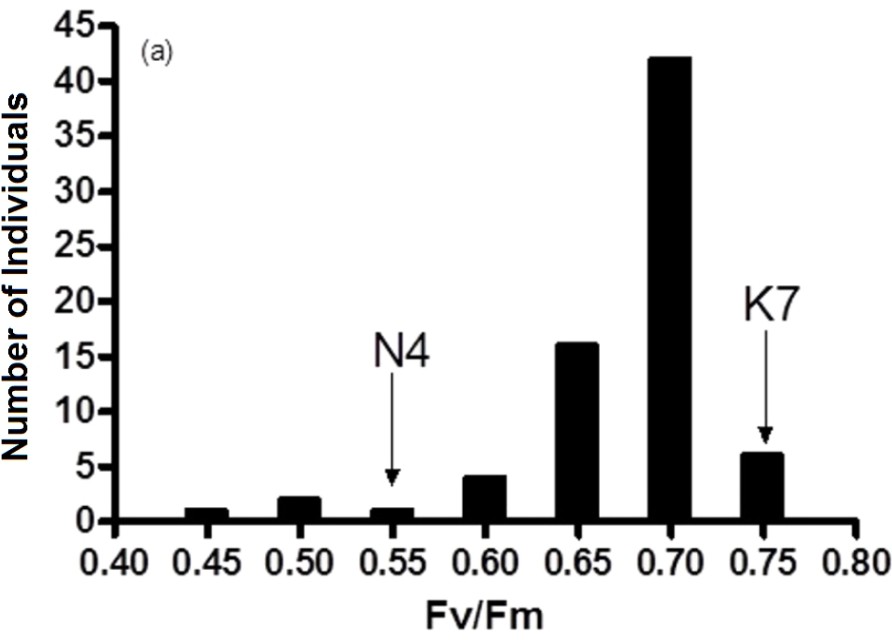

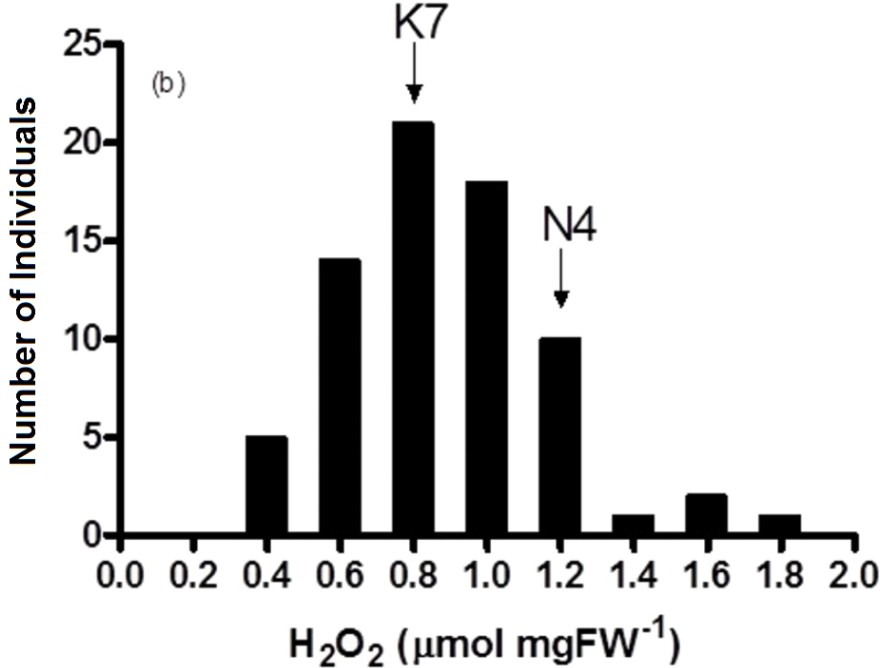

**Figure 4  Frequency distribution of (A) chlorophyll fluorescence (Fv/Fm) and (B) hydrogen peroxide (H₂O₂) for the 72 F1 progeny arrays and the two parental cultivars at 40 days of exposure to heat stress.** (K7) heat tolerant Kangaroo Valley and (N4) heat sensitive Norlea, Each progeny array and each of the two parental cultivars consisted of six replicates.

**Table 2 Analysis of Variance (ANOVA) for chlorophyll fluorescence (Fv/Fm) values and hydrogen peroxide ($H_2O_2$, $\mu$mol mgFW$^{-1}$) content among the 72 F1 progeny arrays at different days of continuous exposure to heat stress.**

| Days of exposure | Range | F value |
|---|---|---|
| Chlorophyll fluorescence (Fv/Fm) | | |
| 0 day | 0.762~0.807 | 2.71*** |
| 10 day | 0.714~0.783 | 2.68*** |
| 20 day | 0.603~0.778 | 3.01*** |
| 30 day | 0.358~0.776 | 4.58*** |
| 40 day | 0.483~0.767 | 25.43*** |
| Hydrogen peroxide ($H_2O_2$) | | |
| 0 day | 0.15~0.52 | 25.76*** |
| 40 day | 0.32~1.74 | 27.95*** |

Notes.
  Each of the two parental cultivars and each progeny array consisted of six replicates.
***The level of statistical significance at $P < 0.001$.

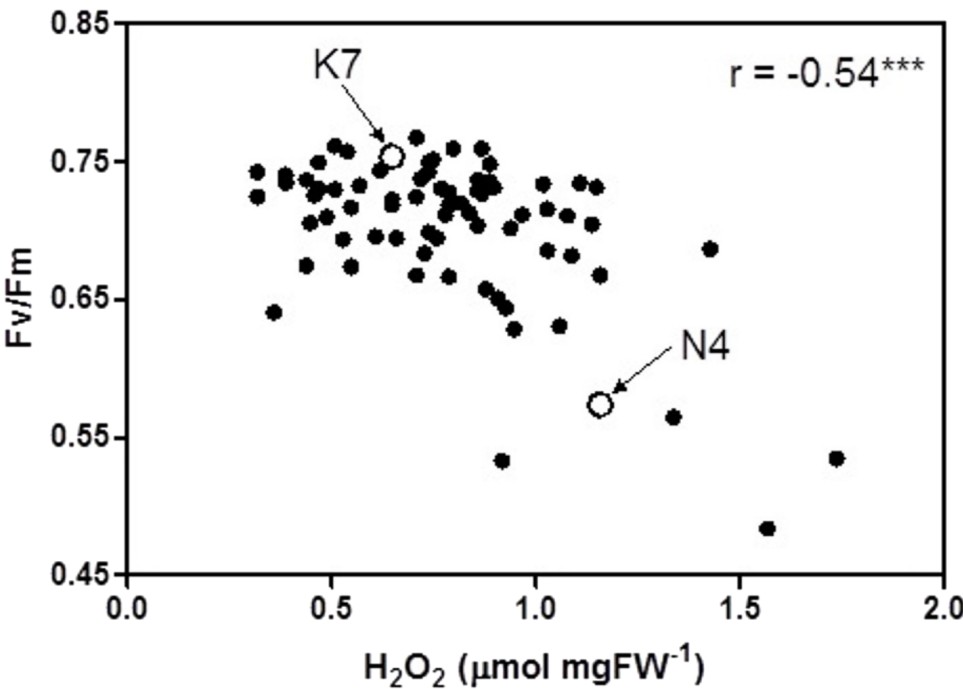

**Figure 5 Correlation between chlorophyll fluorescence (Fv/Fm) and hydrogen peroxide ($H_2O_2$) for the 72 F1 progeny arrays and the two parental cultivars at 40 days of exposure to heat stress.** (K7) heat tolerant Kangaroo Valley and (N4) heat sensitive Norlea. *** indicates the level of statistical significance at $P < 0.001$.

heat sensitive cultivars of perennial ryegrass (*Wehner & Watschke, 1981*; *Jiang & Huang, 2001*; *Zhou & Abaraha, 2007*; *Soliman et al., 2011*; *Soliman et al., 2012*; *Li, Jannasch & Jiang, 2020*). These results show that heat tolerant cultivars of perennial ryegrass had significantly higher photosynthetic performance (higher Fv/Fm values) and lower leaf $H_2O_2$ content,

compared to heat sensitive cultivars. At 40 days of heat stress, approximately 95% of the progeny arrays had Fv/Fm values that were equal to or intermediate to the values of the two parental cultivars and 68% of the progeny arrays had $H_2O_2$ concentrations equal to or intermediate to the two parental cultivars. Conversely, other members of this progeny array have phenotypic trait values beyond their the two parents.

The phenotypic trait distribution for the five leaf growth traits, Fv/Fm values, and $H_2O_2$ content for the 72 progeny arrays is consistent with the distribution expected for traits that are determined by multiple loci (i.e., they are quantitative genetic traits) (*Falconer & Mackay, 1996*). The distribution for these phenotypic traits indicates considerable additive genetic variation among the F1 hybrid progeny arrays, which resulted from crossing the Kangaroo Valley and Norlea cultivars. This additive genetic variation was generated by genetic recombination during gamete formation by the parental plants. In addition, the clustering of many progeny data points near the Kangaroo Valley cultivar, which served as the paternal parent in the cross, may signal the role of dominance (the Kangaroo Valley cultivar possesses dominant alleles) or epistatic interactions in the phenotypic expression of photosynthetic performance and leaf $H_2O_2$ content among the progeny arrays (*Falconer & Mackay, 1996*). Determining the relative contributions of additive genetic variation, and other genetic processes, in the phenotypic expression of the quantitative traits we examined should be the focus of future research.

The photochemical efficiency of PSII, measured by chlorophyll fluorescence (Fv/Fm), is the most sensitive component associated with photosynthesis and it is used commonly to evaluate heat tolerance in plants (*Maxwell & Johnson, 2000*). Under elevated temperatures, ROS are produced through specific metabolic pathways such as photosynthesis and photorespiration (*Queval et al., 2008*). The generation of ROS results from a disrupted balance between photochemical and biochemical reactions inhibiting photosynthesis processes (*Wahid et al., 2007*). Plants however have developed several mechanisms for tolerance to stress such as antioxidant enzymes and HSPs.

The distribution of Fv/Fm and $H_2O_2$ values among the 72 progeny arrays suggests genetic variation for the genes responsible for heat tolerance. These genes control antioxidant activity and the formation of HSPs, which in turn inhibit the formation of ROS and maintain membrane stability and thus increase photosynthetic efficiency, improve plant growth, and allow plants to endure heat stress. These results suggest that the phenotypic variation in heat tolerance exhibited by the 72 progeny arrays analyzed in this study is closely associated with the ability to suppress oxidative stress. This is consistent with previous findings among cultivars of perennial ryegrass (*Soliman et al., 2011*; *Soliman et al., 2012*).

Leaf growth traits also play important roles in plant acclimation to environmental stress (*Terashima et al., 2011*). We did not however detect a relationship between the leaf growth traits we measured prior to the imposition of heat stress and Fv/Fm values and $H_2O_2$ content at 40 days after the imposition of heat stress. Results of the current study differ from those of our previous findings with other perennial ryegrass cultivars (*Soliman et al., 2011*; *Soliman et al., 2012*), which showed significant relationships between leaf traits, especially leaf thickness, and ROS generation and heat tolerance. This discrepancy likely

results from genetic difference of the parental cultivars used in the previous studies. Clearly, heat tolerance is a complex plant response governed by many factors, not least of which is the genetic background of the plants (cultivars) being studied.

## CONCLUSIONS

To the best of our knowledge, this study represents the first assessment of the genetic basis of heat tolerance in perennial ryegrass. This study combined physiological measurements (Fv/Fm and $H_2O_2$ content) within a genetic framework (i.e., parent–offspring comparison) to assess the inheritance of heat tolerance in this grass. Based on the specific crosses used in this study (the Kangaroo Valley and Norlea cultivars), our results indicate considerable additive genetic variation within these progeny arrays. This diversity can be used to improve heat tolerance in cultivars of perennial ryegrass using conventional plant breeding, and could also facilitate marker-assisted breeding, and/or pave the way for characterizing the underlying genetic and genomic factors which could be useful for developing plants with improved heat tolerance (*Sreenivasulu, Sopory & Kishor, 2007*; *Barnabás, Jäger & Fehér, 2008*; *Tricker et al., 2018*).

### Funding

This work was supported by the Deanship of Scientific Research at King Khalid University for funding this work through Research Group Project (Number R.G.P.1/210/41). The funders had no role in study design, data collection and analysis, decision to publish, or preparation of the manuscript.

### Grant Disclosures

The following grant information was disclosed by the authors:
The Deanship of Scientific Research at King Khalid University: Number R.G.P.1/210/41.

### Competing Interests

The authors declare there are no competing interests.

### Author Contributions

- Wagdi S. Soliman conceived and designed the experiments, performed the experiments, analyzed the data, prepared figures and/or tables, authored or reviewed drafts of the paper, and approved the final draft.
- Ahmed M. Abbas analyzed the data, prepared figures and/or tables, authored or reviewed drafts of the paper, and approved the final draft.
- Stephen J. Novak analyzed the data, authored or reviewed drafts of the paper, and approved the final draft.
- Masahiro Fujimori conceived and designed the experiments, performed the experiments, authored or reviewed drafts of the paper, provide the materials, and approved the final draft.

- Kazuhiro Tase conceived and designed the experiments, performed the experiments, prepared figures and/or tables, provide the materials, and approved the final draft.
- Shu-ichi Sugiyama conceived and designed the experiments, performed the experiments, analyzed the data, prepared figures and/or tables, authored or reviewed drafts of the paper, and approved the final draft.

## Data Availability

   The raw measurements are available in the Supplemental File.

## Supplemental Information

Supplemental information for this article can be found online at http://dx.doi.org/10.7717/peerj.11782#supplemental-information.

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
