# Peer review of "Inheritance of heat tolerance in perennial ryegrass (Lolium perenne, Poaceae): evidence from progeny array analysis"

_PeerJ, doi:10.7717/peerj.11782_

## Round 0.1 · original submission · Major Revisions

Although all three reviewers see value in your contribution, substantial improvement in the presentation and discussion of your work is necessary. In particular, you should:
- clearly define your research question/ contribution
- describe your materials (genotypes) and methods in detail
- organize your results and discussion in a congruent way.

Reviewer 1 ·

Basic reporting

The document addresses an important issue for agricultural crops in the context of climate change. It provides information related with the physiology and genetics of adaptation and tolerance to heat stress, using Lolium perenne as a model plant for other C3 agricultural crops. The introduction and discussion of results provides adequate scientific background context. The morphological and physiological traits described in the study were measured using appropriate and sound lab methodology. The manuscript is concise and well written. The article structure is appropriate, in general, but there are some suggestions to improve it, made and marked directly in the document

In the results and discussion section, I would suggest to describe first the results and then discuss them. Thus, I would suggest moving the first three paragraphs of this section to either the introduction (antecedents/context) or after presenting some results. Revision and reorganization of this section would also allow to avoid repetition of some results, as indicated by specific comments in the document.

Experimental design

Several important points are missing or not completely described in the Methods section, particularly those related with the genetic context of the study. This missing information introduces confusion on the genetic interpretation of the results and limits the possibility for replicating the experiment properly. For instance, it is not clear whether the cross between cultivars involved a single-parent cross from each cultivar and the 72 genotypes included in the trial are then full-sibs from a single family. If the cross involved several individuals from each cultivar, then describing the family structure and relationships of the 72 genotypes evaluated would be very useful. Similarly, it is not clear what the authors mean when they mention that “six replications” per genotype were used. How were the genotypes (original parents and the 72 descendant genotypes replicated)? Through their progeny (i.e., sexual propagules) or through clonal replicates (i.e, vegetative propagules). This is an important and crucial issue for interpreting the results, because if the genotypes were actually replicated (either sexually or clonally), then variance components between and within genotypes can be estimated, and, from them, heritability estimates (narrow-sense, broad-sense or clonal repeatability, depending upon the nature of replicates). If so, the authors are missing and excellent opportunity to really address genetic control of traits measured in the study. Otherwise, if genotypes were not really replicated, then the document is dealing only with phenotypic variation within a single full-sib family, and not really evaluating genotypes, nor interpreting genetic variation.

It should also be described how the parents used in the cross were included in the experimental layout with their progeny genotypes. They are not mentioned in the first paragraphs when describing the experiment set-up. Where they evaluated in separate experiments or were they included in the same experimental layout with their progeny. If so, it would be important to mention how they were replicated in order to have similar age and plant size development as their progeny evaluated.

A more detailed description on the leaf sampling procedure for evaluating morphological and physiological traits would also improve interpretation and understanding of the results. For instance, Chlorophyll Fluorescence and H2O2 measurements were done in the same leaves? Same or different leaves were sampled before and after the stress? Number and position of leaves sampled on each plant? How the environmental factors were controlled while sampling and measuring Chl Fluorescence in the 72 genotypes and replications?

Validity of the findings

The results presented in the manuscript are valid, and they seem to be statistically sound. However some doubts on the statistical and genetic interpretation of results arise from the missing information in the methods section about the procedure used for "replicating" the genotypes in the experimental design and trial layout. Clarification on the methodological issues raised would also help to support the conclusions of the study.

Additional comments

In my opinion, the manuscript has valuable information, but in its current stage there are some doubts on the genetic context, interpretation and meaning of results. A revised version addressing the methodological issues mentioned above would substantially improve the quality and meaning of the study.

I have made (and marked) several specific comments, questions, and editing suggestions directly on the PDF file of the manuscript, which should be shared with the authors for their benefit in order to be addressed in a new and improved version.

Annotated reviews are not available for download in order to protect the identity of reviewers who chose to remain anonymous.

Reviewer 2 ·

Basic reporting

Experimental procedures are clear with good acceptable English. Should have provided more references. In few occasion references are missing.

Experimental design

Experimental design is good and fit with the aim and scope of the journal. The research questions are not well defined and no rigorous investigation performed.

Validity of the findings

Nothing new is there about the ROS signaling due to stress responses and hence do not possess any novelty.

Additional comments

The reactive oxygen species stress mediated by H2O2 is a common phenomenon in plant stress biology that lead to decrease in the photosynthetic rate. It impacts the PS II. So there is nothing new in the manuscript. It is a primitive work and do not bear any extra input to the scientific community that can be useful for the reader. Few of the comments are as follows
1. Abstract section: please replace the word “significance”. Else mention the p value.
2. The H2O2 content generally get increased due to stress thereby decreasing the photosynthetic rate. Higher Fv/Fm is related to higher level of photosynthetic activities (PSII). However, during stress conditions, the Fv/Fm ratio get lowered which is explained in the article. However, there is no phenotypic data to show the stress conditions and percentage of stress incurred to the plants.
3. The Fv/Fm value range from 0.79 to 0.84 in normal conditions. However, your Fv/Fm value in stressed plants are lower in the case of N4 and in the suitable range in the case of K7. Why it is in the normal range in the case of K7?
4. Line 154: please explain the statistical details of the significance. This applicable for all parts of the manuscript.
5. Statistical analysis was not conducted for any experiment.

Reviewer 3 ·

Basic reporting

I liked the topic of the manuscript. However, the manuscript needs substantial improvement. All sections, introduction, methodology and result/discussion, all need much work in terms of content and also writing. Latest publications are not cited. The flow in writing is weak, needs more work. Detailed comments are provided "general comments for author" section"

Experimental design

The manuscript needs to state more clearly the importance and scope of this research in relation to the crop under this study which is missing. The research design looks ok to me. However, I have some concerns related to the pot size that was used to grow the plants for more >80 days. Detailed comments are provided "general comments for author" section"

Validity of the findings

No comments.

Additional comments

Overall comments:
The topic of the manuscript is very interesting. But, the manuscript needs further work to improve it. There is a need to add some content, improve the results and discussion section, and further improve the flow in the manuscript.

Abstract section:
General comments:
Needs improvement. The problem in association with the crop should be stated appropriately. The conclusion should include the implications of the study.
Specific comments
Line 24: It may be better to write only “background”
Line 24 to 25: C3 crops including “…”
Line 27-28: Materials and methods need further improvement. Rewrite it.
Lines 29-36: Maybe you can reduce the content on leaf traits to only one sentence considering the results obtained.

Introduction section:
General comments:
The introduction part does not introduce the crop. There should be at least a small paragraph mentioning the importance of the crop and the need for this particular study. It is also important to provide brief importance of key traits evaluated in association with heat stress in the introduction section.
Specific comments:
Lines 50 to 56: Please include recent climate data if possible. The data presented here is good but almost 15 years old.
Line 71: …cereals including common wheat?
Lines 73-75: There are recent studies on heat stress. Please include them and refine this section.

Method section:
General comments:
The language in the method section needs to be improved. The method should be presented with further clarity. Need some improvement in the flow too.
Specific comments:
Line 86: Mention “heat tolerant and susceptible”.
Line 86-87: Need to mention when the cross was made. Similarly, in which generation were the 72 lines derived from the cross?
Line 97: 40 days.
Line 99: Need to define Fv, Fm early on. Fo and Fm are defined later in the paragraph but not Fv.
Lines 118 to 119: Were the plants that were used in taking destructive samples used in the evaluation of parameters of chlorophyll fluorescence kinetics? Not sure but this might have some impact on the assessment.

Results and discussion:
General comments:
The flow of results and discussion need to be improved. I suggest talking about chlorophyll fluorescence in one paragraph, about H2O2 in the other followed by leaf traits. Then, you may want to present the association of these different traits in terms of heat stress tolerance. The discussion part is weak. It needs strengthening with more supporting literature. There may be limited literature available on rye. But similar studies are available for other crops such as wheat.
Specific comments:
Line 140: Are these kinds of studies still limited? If yes, please include a recent citation.
Lines 138 to 153: Some of the information in between these lines may fit better in the introduction section.
Lines 188 to 189: What do you mean by “less genetically controlled”?
Lines 189 to 194: There are many studies that have indicated the importance of leaf traits in conferring heat tolerance. This study did not show that but it is worth discussing in further depth not only to present the importance of leaf traits but also what other reasons could have led to this result.

Conclusion section:
If one of the key findings is that “the traits under study” are genetically controlled, it should be mentioned that there is a possibility of improving heat/thermal tolerance through breeding.

---

## Round 0.2 · Major Revisions

The manuscript has clearly improved. However, 1) please address the comments of the revised version, and 2) provide a point-by-point reply for all comments you resolved (or explain which criticisms you believe are unjustified).

Reviewer 3 ·

Basic reporting

As I indicated earlier, I liked the topic of the manuscript. There is definitely some improvement in the manuscript. One of the weaknesses that still remains in the current version is the language and flow. I still insist on improvement in the focus, especially in the introduction and discussion.

Experimental design

This looks fine to me.

Validity of the findings

No comment.

Additional comments

Abstract:
It looks better but I still insist on some language improvement to make it crispy.
Line 29: Find a better option/word for “great association”.
Line 31: Wide genetic association was “observed”..could be better.

Introduction
I can see some improvement but I see the need to improve the language and flow. The brief importance of traits evaluated in the study is still missing.
I believe that there are recent climate data on heat stress that could be used in the section from line 54-57).
Line 68: Scientific name of wheat.
Line 72-73: This needs recent citation.

Method:
It is improved. But I would suggest one more round of English edit.

Results:
Presentation of results separately looks better.
Some English edits will improve this section.
Line 154-155 is the repetition of information, remove it.

Discussion:
The discussion part looks much improved compared to the previous version. But it needs focus and further strengthening.
Line 174-179 is a kind of repetition from the result section, better remove it.
Lin 171-173: Take it to the paragraph that starts from line 187.

Conclusion
Make the conclusion more succinct. From line 200, the content looks more like a future perspective rather than a conclusion.

---

## Round 0.3 · Major Revisions

Your manuscript has improved, but there are still important issues that need to be resolved:

1) The experimental design is not completely clear; 2) The statistical data interpretation has to be improved.

3) Writing and grammar need to be revised; maybe consider a professional service. Please see the reviewer comments and the annotated PDF for details.

4) The conclusion needs revision as well. The second phrase is much too long and contains a grammar error (plants).

Reviewer 1 ·

Basic reporting

The study was appropriately designed and the morphological and physiological traits described in the study were measured using appropriate and sound lab methodology. The results and discussion is mostly relevant to the specific objectives and hypotheses stated. The manuscript is concise and mostly well written. It has certainly improved in relation to earlier version, but there are a few spelling and gramar errors, that should be considered for correction. They are marked through the document with the tracking tool to facilitate revision and change. The article structure is appropriate, in general, but there are also some suggestions to improve it, made and marked directly in the document. References are all appropriate. Figures and tables are also appropriate, and only some suggestions are made to add or improve information and data provided in them. Suggestions are indicated directly on the Figures and Tables in the PDF file.

Experimental design

The research questions are well defined and the experimental design is appropriatte to those questions; Most suggestions made to the earlier version were appropriately addressed in the revised version, but there are still missing some important points related with the composition and replication of genotypes (parental and 72 F1 full-sib igenotypes involved in the study), and with the statistical model and assumptions considered in the data analysis. Full and detailed description of the genetic materials evaluated in the study, the precise number of replications (individual plants) for each genotype used in the trial and the consideration of "fixed-effects" or random-effects" model for genotypes is required in order to be able to appropriately replicate the experiment and would help to interpret correctly the results obtained.

Considering the information provided in the revised version of manuscript, it seems that parental and 72 F1 full-sib genotypes obtained from a single cross between the two parental genotypes were tested by using 6 (or 12?) plants from each genotype (i.e, F2 progenies from the F1 full sibs?) but the question is how these progenies were obtained? by self- or open-pollination?

In addition, the clarification of the "fixed-effects" or "random-effects" model in the statistical analyses of data is required to avoid confusion by using ambiguously the terms "significant differences" or "significant variation". Additional specific comments are included directly in the Word and PDF files of the revised version of manuscript.

Validity of the findings

Results presented in the revised manuscript are valid, and they are statistically sound, provided the doubt raised about the number of replications used for each single genotype (particularly for the 72 F1 full-sibs) is properly solved. Clarification on the methodological issues mntioned before and commented on the manuscript would solve these issues and properly support the conclusions. However, the terms and concepts "significant differences" vs. "significant variation", and "genetic variation" vs. "phenotypic variation" should be unambiguously used, once the replication and model assumptions issues are clarified and solved..

Additional comments

In summary, in my opinion, the manuscript has clearly improved from the earlier version, but there still are a few and important issues and metodology questions that need to be addressed and clarified, providing additional details, in order to reach its full potential and meet quality standards for publication in this journal.

I have made (and marked) several specific comments, questions, and editing suggestions directly on the Word and PDF files to help the authors the specific questions and doubts that require clarification. The edit-tracking tools used to identify these comments and suggestions would facilitate to carry out the revision, and obtain a new, improved version.

Please, check comments made on both files, particularly those in the PDF file for Tables and Figures

Annotated reviews are not available for download in order to protect the identity of reviewers who chose to remain anonymous.

---

## Round 0.4 · Major Revisions

The writing has improved substantially. However, you should read again the paper before resubmission. Some of the changes need validation, for example 'Please check this value, I believe the percentage is greater than 72%.'; this value was modified to be 90 percent! There are also funny typos in your rebuttal letter ("reputtal"; "satirical analysis").
Besides a carefull preparation of your revision, you should justify clearly the authorship of Stephen Novak [contribution and authorship criteria]. For your next manuscript version I will invite also new reviewers.

Reviewer 1 ·

Basic reporting

The document structure and writing has improved substantially in the revised version. There are only a few spelling/typing errors, which are identified and marked in the revision file attached. These typing errors include some citations in which the author’s name or the publication year doesn´t coincide with the references listed.

Most of the comments made to earlier versions of the manuscript have been addressed by the authors in the newly revised version. However, there are still a couple of basic issues in the experimental protocol which haven´t been clarified. These two issues are important because they create confusion when interpreting the results obtained. Moreover, the information provided about these two issues in the earlier versions of the manuscript have been contradictory, adding to this inconsistency and confuse use of basic statistical principles supporting the proper interpretation of results. The authors mention that “seeds of the two parental cultivars and the progeny were germinated” and later on, they also mention using a “randomized complete block experimental design”, but they never mention how many seeds or replications were used for each parental cultivar and the progeny. Furthermore, when they describe the statistical analysis they mention that they “tested for significant differences between the two parental cultivars and among the progeny, implying that both the parental cultivars and the individual progeny were replicated. But they never describe how they obtained and used the replicas of each individual progeny.

In this sense, if the parental cultivars and the progeny were replicated, to be consistent with the description of the statistical analysis and the “F values” provided in Tables 1 and 2, they should address and clarify the questions of how many replicates were used in each case (for parental cultivars and for each individual in the progeny) and how the replicates for the individuals in the F1 progeny were produced, as requested in earlier revisions.

On the other hand, if the parental cultivars or the individuals in the progeny were not replicated (if only a sample of 72 individuals in the progeny were evaluated), then the experimental design should be described differently, since in such a scenario it is not possible to “test for significant differences among the progeny”, and the meaning of the F value in Tables 1 and 2 should be explained properly, as well as the interpretation of results shown in those Tables. In this scenario, only the basic descriptive statistics (mean, variance, standard deviation, range of extreme values, ec.) for the progeny sample can be obtained and described. The variance of the sample would be an estimate of the phenotypic variance among the progeny for the traits measured in the study.
.
I have made (and marked) several specific comments, questions, and editing suggestions directly on the Word and PDF files to help the authors the specific questions and doubts that require clarification. The edit-tracking tools used to identify these comments and suggestions would facilitate to carry out the revision, and obtain the final version.

Experimental design

Already addressed in the first section (and in earlier revisions)

Validity of the findings

Already addressed in the first section (and in earlier revisions)

Additional comments

Please address carefully and thoroughly all the specific comments, questios and suggestions marked directly in the attached Word/PDF File containing the revised manuscript.

Annotated reviews are not available for download in order to protect the identity of reviewers who chose to remain anonymous.

---

## Round 0.5 · Minor Revisions

Your manuscript has improved a lot. There are only some minor comments of reviewer 4 that need to be addressed.

Reviewer 1 ·

Basic reporting

The current, revised version of the manuscript has addressed all the concerns and issues raised in the earlier versions. The authors have properly answered and included all the suggestions made by the reviewer. Thus, the revised version of the manuscript could be accepted for publication. There are only a couple of spelling/typing errors, which are indicated and edited directly on the manuscript using the tracking tools, so they can be solved by the authors or directly by the editor.

Experimental design

No comment

Validity of the findings

No comment

Additional comments

No comment

Annotated reviews are not available for download in order to protect the identity of reviewers who chose to remain anonymous.

·

Basic reporting

Some of the figure legends (1, 2, 3 and 4) appear to be incomplete

Experimental design

No comment

Validity of the findings

No comment

Additional comments

In my opinion the manuscript ‘Inheritance of heat tolerance in perennial ryegrass (Lolium perenne, Poaceae): evidence from progeny array analysis’ is a clearly presented report of a robustly designed experiment considering an important aspect of forage selection. The results support the conclusions and, if implemented, the advances in plant breeding and selection will ensure that perennial ryegrass continues to be a productive component of pasture based livestock production systems in the future. I only have a few minor comments for the authors to address:
Line 120: change ‘temperate climates regions’ to ‘temperate climatic regions’
Line 138 - 139: change ‘they had higher’ to ‘the plants had higher’
Line 185: please clarify the N-P-K ratio
Line 234: change ‘for significance differences’ to ‘for significant differences’
Line 247: change ‘All statistical analysis’ to ‘All statistical analyses’
Line 256: please add ‘and table 1’ to Fig. 3 in parentheses as this shows that there were significant differences
Line 260 – 261: I am not convinced that figure 3d shows that the leaf growth traits of the majority of the progeny were equal to or greater than the values of the two parental cultivars and suggest that it is removed
Line 346: change ‘we measure prior’ to ‘we measured prior’

---

## Round 0.6 · accepted · Accept

Thank you for addressing the additional comments in your revised manuscript!